# Association between the Maternal Gut Microbiome and Macrosomia

**DOI:** 10.3390/biology13080570

**Published:** 2024-07-28

**Authors:** Zixin Zhong, Rongjing An, Shujuan Ma, Na Zhang, Xian Zhang, Lizhang Chen, Xinrui Wu, Huijun Lin, Tianyu Xiang, Hongzhuan Tan, Mengshi Chen

**Affiliations:** 1Hunan Provincial Key Laboratory of Clinical Epidemiology, Xiangya School of Public Health, Central South University, Changsha 410013, China; 13574652893@163.com (Z.Z.); msj1008@163.com (S.M.); xingxinghongwei@gmail.com (N.Z.); chenliz@csu.edu.cn (L.C.); linhuijun9586@163.com (H.L.); xiangtianyu163@163.com (T.X.); tanhz@mail.csu.edu.cn (H.T.); 2Department of Epidemiology and Health Statistics, Xiangya School of Public Health, Central South University, Changsha 410013, China; 3Chaoyang District Center for Diseases Prevention and Control of Beijing, Beijing 100020, China; an1503416@163.com; 4Reproductive and Genetic Hospital of CITIC-Xiangya, Clinical Research Center for Reproduction and Genetics in Hunan Province, Changsha 410008, China; 5Department of Occupational and Environment Health, Xiangya School of Public Health, Central South University, Changsha 410013, China; zixuange2010@126.com; 6School of Medicine, Jishou University, Jishou 416000, China; wuxinrui0605@163.com

**Keywords:** pregnant, macrosomia, gut microbiome, metagenomic analysis, biomarkers

## Abstract

**Simple Summary:**

Fetal macrosomia is when a baby’s weight at birth is equal to or greater than 4000 g or 4500 g. The rising incidence of macrosomia poses a significant challenge in obstetrics, as it can have serious health consequences for both mothers and babies. The maternal gut microbiome can influence the health of pregnant women and their babies, with potential effects on birth weight. However, research on the link between the microbiome and birth weight, especially macrosomia, is limited; further investigation is needed. Here, we discovered a connection between the maternal gut microbiome and macrosomia. Our findings present novel opportunities for preventing macrosomia by manipulating the composition of the intestinal microbiota. Early prediction models using gut microbiota and clinical indicators can predict macrosomia.

**Abstract:**

Fetal macrosomia is defined as a birthweight ≥4000 g and causes harm to pregnant women and fetuses. Studies reported that the maternal intestinal microbiome plays a key role in the establishment, growth, and development of the fetal intestinal microbiome. However, whether there is a relationship between maternal gut microbiota and macrosomia remains unclear. Our study aimed to identify gut microbiota that may be related to the occurrence of macrosomia, explore the possible mechanisms by which it causes macrosomia, and establish a prediction model to determine the feasibility of predicting macrosomia by early maternal gut microbiota. We conducted a nested case-control study based on an early pregnancy cohort (ChiCTR1900020652) in the Maternity and Child Health Hospital of Hunan Province on fecal samples of 93 women (31 delivered macrosomia as the case group and 62 delivered normal birth weight newborns as the control group) collected and included in this study. We performed metagenomic analysis to compare the composition and function of the gut microbiome between cases and controls. Correlation analysis was used to explore the association of differential species and differential functional pathways. A random forest model was used to construct an early pregnancy prediction model for macrosomia. At the species level, there were more *Bacteroides salyersiae*, *Bacteroides plebeius*, *Ruminococcus lactaris*, and *Bacteroides ovatus* in the intestinal microbiome of macrosomias’ mothers compared with mothers bearing fetuses that had normal birth weight. Functional pathways of the gut microbiome including gondoate biosynthesis, L-histidine degradation III, cis-vaccenate biosynthesis, L-arginine biosynthesis III, tRNA processing, and mannitol cycle, which were more abundant in the macrosomia group. Significant correlations were found between species and functional pathways. *Bacteroides plebeius* was significantly associated with the pathway of cis-vaccenate biosynthesis (r = 0.28, *p* = 0.005) and gondoate biosynthesis (r = 0.28, *p* < 0.001) and *Bacteroides ovatus* was positively associated with the pathway of cis-vaccenate biosynthesis (r = 0.29, *p* = 0.005) and gondoate biosynthesis (r = 0.32, *p* = 0.002). *Bacteroides salyersiae* was significantly associated with the pathway of cis-vaccenate biosynthesis (r = 0.24, *p* = 0.018), gondoate biosynthesis (r = 0.31, *p* = 0.003), and L–histidine degradation III (r = 0.22, *p* = 0.291). Finally, four differential species and four clinical indicators were included in the random forest model for predicting macrosomia. The areas under the working characteristic curves of the training and validation sets were 0.935 (95% CI: 0.851~0.979) and 0.909 (95% CI: 0.679~0.992), respectively. Maternal gut microbiota in early pregnancy may play an important role in the development of macrosomia and can be used as potential predictors to prevent macrosomia.

## 1. Introduction

Fetal macrosomia is defined as any gestational age with fetal weight that reaches or exceeds absolute birth weight (usually 4000 g or 4500 g) [1]. In recent years, the increase in the incidence of macrosomia has become a major challenge in obstetrics. According to the National Center for Health Statistics, approximately 7.80% of newborns in the United States weigh more than 4000 g. In China, the incidence of macrosomia has increased from 6.00% to 8.70% over 20 years [2]. Macrosomia can cause great harm to the health of mothers and babies. During delivery, a large fetus often leads to intrauterine distress, neonatal asphyxia, shoulder dystocia, and brachial plexus injury [3]. For pregnant women, the excessive head diameter of the fetus leads to asymmetry of the head and pelvis, which may cause premature rupture of membranes, prolonged labor, complete or partial uterine rupture, and, in severe cases, endangering the lives of the mother and infant [4]. Moreover, macrosomia also predicts being overweight and obese in children and is associated with type 1 or 2 diabetes in adulthood as well as cardiovascular disease [5,6]. Several risk factors such as excessive gestational weight gain (GWG), past history of macrosomia, pre-pregnancy care, and gestational diabetes mellitus (GDM) lead to high birth weight [7,8]. Moreover, placental function is also an important factor affecting fetal growth and development [9] and the expression of proteins related to glucose and fatty acid transport in placental tissue is closely related to the occurrence of macrosomia.

The human gut microbiome is a large and complex micro-ecosystem in the body, containing trillions of microbial cells [10]. The gut microbiome can be regarded as a “microbial organ” that regulates food digestion, energy utilization [11], neurotransmitters [12], drug metabolism [13], and host metabolism and its function is closely related to host physiology and pathophysiology [14]. The maternal gut microbiome, one of the major microbial reservoirs in humans and a source of the gut microbiome in early infancy [15], may also influence the health of pregnant women and their offspring, although the exact mechanisms have not been fully defined [16]. A study reported that the composition and function of the gut microbiome before and during pregnancy may influence birth weight [17]. Specifically, a positive correlation was found between maternal *E. coli* and the birth weight of the baby [18]. The gut microbiome of pregnant women can also affect several metabolic activities during pregnancy that affect metabolism such as increased carbohydrate metabolism and butyrate production [19,20]. These functions may influence fetal growth by affecting the maternal nutritional status and GWG [17]. Moreover, the intestinal microbiome composition of the mother may influence pregnancy outcomes by affecting the maternal adaptation to pregnancy, placental function, and the environment of the fetus in the mother’s body [21]. However, previous studies have focused on the influence of the microbiome on preterm. The correlation between intestinal microbiome and birth weight, especially of macrosomia, is largely unknown and more specific research is needed to explore the relationship between them.

Here, we aimed to evaluate the characteristics of the gut microbiome in pregnant women with macrosomia and investigate whether there is a relationship between the composition and function of the maternal gut microbiome and fetal macrosomia by using the metagenomic sequence data from an early pregnancy cohort study in China.

## 2. Methods

### 2.1. Study Design

Participants were selected from an early pregnancy follow-up cohort at the Maternity and Child Health Hospital of Hunan Province, China. This cohort was established from Mar 2017 to 2018 and it was approved by the Medical Ethics Committee of Hunan Maternal and Child Health Hospital (No. EC201624). All pregnant women participating in the study signed informed consent forms. The inclusion criteria of this cohort were as follows: single birth; natural conception; no history of diabetes, hypertension, thyroid disease, cardiovascular and cerebrovascular diseases before pregnancy; no acute infection in the past two weeks; did not use drugs that may affect glucose metabolism; and women at early pregnancy (10–13^+6^ weeks) who plan to complete the checkup and delivery at the selected research site. Participants were recruited in the first trimester (10–14 weeks) and followed up to 42 days postpartum. In the end, a total of 870 participants were included in the early pregnancy cohort and 820 participants were followed up from early pregnancy to postpartum. The dropout rate in this cohort was 6.0%. We conducted a nested case-control study based on this cohort, with inclusion criteria for cases: delivery of macrosomia (birth weight ≥ 4000 g), complete questionnaire and blood sample data, and pregnant women who did not take probiotics during pregnancy. A total of 38 pregnant women gave birth to macrosomia, with an incidence rate of 5.1%. After excluding pregnant women with significant missing clinical data, a total of 31 pregnant women who delivered macrosomia were included in the case group. The control group randomly selected pregnant women in the same cohort with a birth weight between 2500 g and 4000 g, complete questionnaire and blood sample data, and without taking probiotics during pregnancy, in a 1:2 ratio based on age (±3 years old) and stool/blood sample collection time (±1 month). In the end, 31 were pregnant with macrosomia as cases and 62 were pregnant with normal birth weight newborns as controls. The blood samples were used for biochemical testing. The stool samples were subjected to shotgun metagenomic sequencing. Moreover, the clinical parameters of the 93 mothers in the first trimester were collected and the birth weight of their infants was recorded.

### 2.2. Stool Sample Processing

Stool samples were independently collected from pregnant women using a sterile stool collector and stored in a −80 °C low-temperature freezer. The total nucleic acids were extracted from 180–200 mg feces using the QIAamp Fast DNA Stool Mini Kit (Qiagen, Hilden, Germany). The metagenomic sequencing was performed by Beijing Nuohe Zhiyuan Bio-Information Technology Co., Ltd. Briefly (Beijing, China), the DNA concentration was measured using the Qubit dsDNA Assay Kit in Qubit 2.0 Flurometer (Life Technologies, Carlsbad, CA, USA). A total amount of 1 μg DNA per sample was used as the input material for the DNA sample preparations. Sequencing libraries were generated using NEBNext Ultra™DNA Library Prep Kit for Illumina (NEB, Ipswich, MA, USA) following the manufacturer’s recommendations and index codes were added to attribute sequences to each sample. The clustering of the index-coded samples was performed on a cBot Cluster Generation System. After cluster generation, the library preparations were sequenced on an Illumina HiSeq platform and paired-end reads were generated. Preprocessing of the Raw Data obtained from the Illumina HiSeq sequencing platform (Illumina, San Diego, CA, USA) using Readfq (V8, https://github.com/lh3/readfq (15 July 2022)) was conducted to acquire the clean data for subsequent analysis. The range of metagenomic reads kept after quality control for each sample was shown in Appendix A.

### 2.3. Metagenomic Data Processing

Metaphlan2 was used to characterize the taxonomic profiles of whole-metagenome shotgun (WMS) samples and was used successfully in large-scale microbial community studies. MetaPhlAn2 relies on unique clade-specific marker genes identified from ~17,000 reference genomes (~13,500 bacterial and archaeal, ~3500 viral, and ~110 eukaryotic). It compares the sequencing reads with the marker and determines the species category; the number of each species in the alignment will be calculated with the marker length content [22]. Taxonomy abundances were calculated and compared between cases and controls. HUMAnN2 was used to investigate the differential function of gut microbiome between the two groups. It is a new approach for functional profiling that aims to describe the metabolic potential of a microbial community and its members, which used MiniPath, Metacyc, and Uniref to efficiently and accurately identify the presence/absence and abundance of microbial pathways in a community from metagenomic or metatranscriptomic sequencing data such as millions of short DNA/RNA reads [23]. Pathway abundances were calculated and compared between the cases and controls. 

### 2.4. Diversity Analysis

Alpha diversity was calculated based on the species level to find differences in the microbiome. Beta diversity was used to discover differences in taxonomies and functional pathways. Alpha diversity was represented by the Simpson index and Shannon index which were calculated from the Metaphlan2 output using the Vegan packages in R. Differences in the alpha diversity between the cases and controls were assessed by using a Student’s t-test and Kruskal–Wallis test in R. Beta diversity was represented by Bray–Curtis distances, which were performed using a PERMANOVA test (Adonis function in vegan) and visualized with a principal coordinate analysis (PCoA) plot using ggplot2 packages.

### 2.5. Statistical Analysis

Continuous variables were described as mean ± standard deviation (SD) if they were normally distributed or median (interquartile range, IQR) if not normally distributed. Enumeration variables were described by rate and proportion. Two sample t-tests, the chi-squared test, and the Mann–Whitney U test were used to compare the differences in characteristics between the two groups that were identified. We performed multivariate linear regression analysis (MaAsLin2), which implements linear mixed-effects models that were useful for multivariable association discovery in population-scale microbiome studies to find a differential abundance of taxonomies and functional pathways; a *p* value < 0.05 and corrected *p*-value (Q value) < 0.25 was considered as a statistical difference. Spearman rank analysis was used to find a correlation between differential taxonomies and differential pathways in R. *p* < 0.05 (Fdr adjusted) was considered a significant correlation. The Corrplot package was used to visualize these correlations. All samples were randomly divided into two parts, the training set and validation set, and the samples in the training set were analyzed by a random forest package of R on the basis of the differential species in the case group and the control group and the differential clinical indicators obtained in this study. We also constructed a random forest classifier on the training set and validated it on the validation set samples. The receiver operating characteristic (ROC) curve analysis of the model was performed using the pROC package of the R software (4.3.1).

## 3. Results

### 3.1. Characteristics of the Participants

The study population’s characteristics were detailed in Table 1, showing higher waist circumference (WC) (82.28 vs. 77.35), weight (58.69 vs. 52.31), and body mass index (BMI) (22.87 vs. 20.77) in the case group compared to the control group, with statistically significant variations. There were no statistically significant differences in the history of diseases such as GDM, Gestational Hypertension, and Adverse Pregnancy between the two groups and all of the participants had no history of GDM. For the mothers’ biochemical parameters in the first trimester, significant differences between cases and controls were found in albumin (44.79 vs. 45.98), triglycerides (TGs) (1.76 vs. 1.47), and total cholesterol (4.89 vs. 4.49). As for eating habits, daily eggs (118.03 vs. 81.89), peanuts (134.33 vs. 93.71), and salt (11.90 vs. 10.93) intake in cases were higher than controls. Moreover, 77.40% of participants who gave birth to macrosomia opted for cesarean section as the mode of delivery. No statistically significant variances were observed in obstetric history, including Gravidity, Parity, and Abortion, between the two groups. Additionally, there were no statistically significant differences in the sex and premature labor of newborns. However, the birth weight of newborns in the case group exceeded that of the control group, with statistically significant variations. 

### 3.2. Comparison of Different Bacteria

The metagenomic analysis of stool samples identified the presence of 9 phyla, 17 classes, 23 orders, 32 families, 123 genera, and 316 species. At the phylum level, Bacteroidetes and Firmicutes were found to be the predominant phyla. We randomly selected the number of observed species detected at different sequencing depths to construct a rarefaction curve and the results showed that as the sequencing depth increased, the curve plateaued, indicating that the amount of sequencing data was reasonable. Further increasing the sequencing amount will reduce the marginal contribution to discovering new observed species (Figure 1A). The comparison of gut microbiome relative abundance at the phylum level between cases and controls is illustrated in Figure 1B. To assess disparities in the maternal intestinal microbiome composition between the two groups, the Simpson index and Shannon index were utilized to measure α diversity. For the gut microbiome of pregnant women, there were no significant differences between the two groups in the Simpson index and Shannon index and we showed the plot in Appendix A. Beta diversity was expressed by the Bray–Curtis distance and the Adonis test showed that there were no significant differences between the two groups. The PCoA of species level at the gut microbiome also showed no distinction between cases and controls in both the PCoA1 and PCoA2 (Appendix A). To investigate whether some of the gut microbiome of pregnant women is associated with the occurrence of macrosomia, the gut microbiome at the species level of the case and control groups were compared using MaAsLin2. At the species level, there were more *Bacteroides salyersiae* (*p* < 0.001, Q < 0.001), *Bacteroides plebeius* (*p* < 0.001, Q < 0.001), *Ruminococcus lactaris* (*p* < 0.001, Q < 0.001), and *Bacteroides ovatus* (*p* = 0.003, Q = 0.127) in cases. After adjusting for eating habits including the average daily intake of salt and peanut, BMI, and waist, these species still have differences between the two groups. The results are shown in Table 2.

### 3.3. Functional Analysis

Functional profiling was performed using HUMAnN2 and the comparisons of pathway abundances were analyzed by MaAsLin2. There were seven pathways significantly different between the two groups. They were the mannitol cycle (*p* < 0.001, Q = 0.002), L-arginine biosynthesis III (*p* < 0.001, Q = 0.011), tRNA processing (*p* < 0.001, Q < 0.011), L-histidine degradation III (*p* < 0.001, Q < 0.014), gondoate biosynthesis (*p* < 0.002, Q < 0.131), cis-vaccenate biosynthesis (*p* < 0.004, Q < 0.223), and thiamin salvage II (*p* = 0.005, Q = 0.231). After adjusting for WC, BMI, and eating habits by MaAsLin2, thiamin salvage II was not significantly different between these two groups, just six pathways were significantly different between the two groups and they were enriched in the cases group. All of these results are in Table 3. 

### 3.4. Correlation Analysis

In our investigation, we examined the correlation between various functional pathways and different species in cases. Our findings indicate that *Bacteroides plebeius* exhibited a significant association with the pathway of cis-vaccenate biosynthesis (r = 0.28, *p* = 0.005) and gondoate biosynthesis (r = 0.28, *p <* 0.001). Similarly, *Bacteroides ovatus* showed a significant association with the pathway of cis-vaccenate biosynthesis (r = 0.29, *p* = 0.005) and gondoate biosynthesis (r = 0.32, *p* = 0.002). Additionally, *Bacteroides salyersiae* displayed a significant association with the pathway of cis-vaccenate biosynthesis (r = 0.24, *p* = 0.018), gondoate biosynthesis (r = 0.31, *p* = 0.003), and L–histidine degradation III (r = 0.22, *p* = 0.291). The results are shown in Figure 2.

### 3.5. Construction of Predictive Models

A random forest algorithm was utilized to construct a predictive model for macrosomia, with 80% of the subjects allocated to the training set and the remaining 20% to the validation set for model rehearsal to select the optimal set. The predictive model for macrosomia was initially constructed using the significantly different species *Bacteroides salyersiae*, *Bacteroides plebeius*, *Ruminococcus lactaris,* and *Bacteroides ovatus* as predictors. The ROC curve results showed that the AUC (Area Under Curve) of the training set was 0.907 (95% CI: 0.816~0.963), the sensitivity was 86.36%, and the specificity was 94.12%. The validation set AUC was 0.864 (95% CI: 0.622~0.977), the sensitivity was 88.89%, and the specificity was 88.89%. The importance analysis was conducted on the indicators included in the model based on the average descent accuracy and average descent Gini coefficient. The results showed that the order of importance ranking was *Ruminococcus lactaris*, *Bacteroides plebeius*, *Bacteroides salyersiae,* and *Bacteroides ovatus*. The results are shown in Figure 3A–C. Then, we added significant clinical indicators in Table 1 including the waist, BMI, TG, TC, and ALB in this model and the the ROC curve results showed that the AUC of the training set was 0.935 (95% CI: 0.851~0.979), the sensitivity was 82.61%, and the specificity was 93.88%. The validation set AUC was 0.909 (95% CI: 0.679~0.992), the sensitivity was 85.71%, and the specificity was 81.82%. The order of importance ranking showed that *Ruminococcus lactaris*, *Bacteroides plebeius,* and *Bacteroides salyersiae* were more important than clinical indicators; the results are shown in Figure 4A–C.

## 4. Discussion

In this study, we characterized the maternal intestinal microbiome associated with fetal macrosomia using shotgun metagenomics. Compared to 16sRNA, the metagenome was able to provide comprehensive information about the gut microbiome and directly identify functional pathways. To date, a limited number of studies on the relationship between the maternal gut microbiome and macrosomia are available. Our study identified new gut microbiota that are different from previous studies and further explored the impact of these microbiota and their functional roles on the occurrence of macrosomia. Based on clinical indicators and gut microbiota, we constructed a new prediction model for macrosomia, providing new ideas for the prevention of macrosomia. This study provided a basis for exploring the relationship between maternal gut microbiota and macrosomia and identified gut microbiota that may be used for early diagnosis and prediction of macrosomia.

### 4.1. Gut Microbiota That Affect the Occurrence of Macrosomia

For the overall composition of gut microbiota, no significant differences were found in alpha and beta diversity between the case group and the control group. Ethan et al. also found that the Shannon index of gut microbiota in mothers is not related to the birth weight of infants [17]. However, in some specific bacterial species composition, we found differences between these two groups. The research results analyzed by Maaslin2 showed that, after adjusting for confounding factors such as BMI, waist, and dietary habits, *Bacteroides salyersiae*, *Bacteroides plebeius*, *Ruminococcus lactaris,* and *Bacteroides ovatus* still showed significant differences between the two groups and they were all enriched in the case group. *Bacteroides ovatus* was found to be increased in obese patients and positively correlated with impaired glucose tolerance [24]. A study about T2D showed that *Bacteroides ovatus* is the most common strain in T2D patients. Its mechanism on blood glucose metabolism may be due to the damage of the intestinal barrier. *Bacteroides ovatus* crosses the damaged intestinal barrier and enters the systemic circulation, while too much *Bacteroides ovatus* in the peripheral blood will drive the expansion of MAIT cells in the body, thus producing more IL-17, which can induce pro-inflammatory reactions in the body through various pathways, thereby inducing insulin resistance and increasing the risk of developing T2D. *Bacteroides plebeius* was found to be enriched in patients with diabetic kidney disease, IgA nephropathy, and hypertension [25,26,27], suggesting that this species is related to human metabolism. In the present study, no association has been found between *Bacteroides salyseriae* and the diseases. Moreover, *Bacteroides salyersiae*, *Bacteroides plebeius,* and *Bacteroides ovatus* all belong to the genus *Bacteroides*, which has been found to be correlated with some metabolic diseases during pregnancy such as GDM and increased GWG. A previous study also reported that *Bacteroides* were significantly positively correlated with glucose [28] and they reported that the possible mechanism was that *Bacteroidetes* mainly produce two short-chain fatty acids (SCFAs): propionate and acetate [29]. Studies have shown that dietary intake of propionate in humans can lead to increased postprandial plasma concentrations of glucagon, fatty acid-binding protein 4 (FABP4), and norepinephrine, which may cause insulin resistance and then increase glucose [30]. Acetate is obesogenic. Animal experiments found that acetate can stimulate parasympathetic activity to increase ghrelin secretion and glucose-stimulated insulin secretion, thereby promoting overeating, hypertriglyceridemia, and increased liver and muscle fat storage [31]. More fat accumulation and glucose can affect the maternal metabolic status, leading to excessive blood lipids and glucose in pregnant women. Excessive TGs and glucose will increase fetal insulin secretion and then stimulate insulin-like growth factor (IGF)-1 secretion in the fetus [32,33], thereby affecting fetal growth and development. Therefore, considering the role of these differential species in regulating blood glucose and lipid levels in the human body, it can be speculated that when these species are enriched in pregnant women, it may increase blood glucose and lipid levels, causing disorders in maternal glucose and lipid metabolism, resulting in excessive blood glucose and lipid transport to the fetal body, and leading to the occurrence of macrosomia. In addition, *Ruminococcus lactaris* was also enriched in case groups. *Ruminococcus lactaris* belongs to the genus *Ruminococcus*. Cortez et al. found that in comparison to healthy pregnant women, GDM patients had a higher abundance of *Ruminococcus* [34]. Moreover, more studies reported that *Ruminococcus* was more abundant in obese adults [35,36,37]. *Ruminococcus* belongs to the family Ruminococcaceae, which is involved in energy metabolism, insulin signaling, and inflammatory processes, and an increase in the relative abundance of Ruminococcaceae correlated with fasting glucose concentration and IR led to a greater risk of GDM development [38]. Therefore, *Ruminococcus lactaris* may increase the risk of macrosomia by increasing gestational weight or participating in GDM.

### 4.2. The Functional Pathways of Gut Microbiota and Their Association with Macrosomia 

Furthermore, we used Maaslin2 and found that the mannitol cycle, L-arginine biosynthesis III, tRNA processing, L-histidine degradation III, gondoate biosynthesis, and cis-vaccenate biosynthesis were all enriched in the cases group. Gondoate biosynthesis involves the synthesis and elongation of long-chain fatty acids (LCFAs) [39]. LCFAs regulate energy metabolism and are involved in the synthesis of triglycerides [40]. Thus, excessive LCFA uptake may lead to triglyceride deposition. This pathway was related to the metabolism of human lipids. Meanwhile, cis-vaccenate is mainly involved in the synthesis of monounsaturated fatty acids (MUFAs), which are also the main component of TGs (priority stored fatty acids) [41]. When these two pathways increase the synthesis of fatty acids, it will lead to an increase in TG [42]. The excessive storage of TG not only triggers systemic metabolic inflammation but also promotes fat breakdown, leading to a decrease in insulin sensitivity and thus affecting glucose metabolism [43]. Therefore, it can be inferred that both the biosynthesis of gondoate and cis-vaccenate in the gut microbiota may cause the aforementioned metabolic changes in pregnant women, leading to the occurrence of macrosomia. Previous studies have found that the L-histidine degradation pathway of gut microbiota is associated with the occurrence of thyroid nodules, Alzheimer’s disease, and coronary heart disease [44,45,46]. However, the specific mechanism is not clear. L-histidine is a nutritional component of the human body and an important functional amino acid. It has various physiological functions such as immune regulation and, antioxidant and anti-inflammatory properties; furthermore, the metabolic changes in amino acids are related to the enhancement of lipid metabolism [47,48]. Therefore, the degradation of L-histidine is likely to affect lipid metabolism in the mother’s body, thereby affecting the growth and development of the fetus. L-arginine is a substrate for the family of nitric oxide (NO) synthase (NOS) enzymes that generate NO, a key chemical involved in normal endothelial function and, hence, cardiovascular health [49]. Previous studies reported that L-arginine may have beneficial effects on carbohydrate and lipid metabolism. However, some studies involving diseased subjects and prospective studies with healthy people found that a higher level of L-arginine is associated with the worsening of an existing disease or may be a potential risk factor for the development of some diseases (CHD, kidney disease, T2DM) [50]. Therefore, it is not yet clear as to how this pathway plays a role in the occurrence of macrosomia in the case group. The pathway of tRNA processing and the mannitol cycle were also enriched in cases; however, the mechanism underlying these observations and its impact on human health needs further investigation.

### 4.3. Correlation Analysis between Differential Species and Different Pathways

*Bacteroides salyersiae*, *Bacteroides plebeius*, and *Bacteroides ovatus* were all significantly related to the pathway of gondoate biosynthesis and cis–vaccenate biosynthesis, both of which are more abundant in cases and involved in the metabolism of lipids in humans. Therefore, these species may affect maternal blood lipids through the above-mentioned functions. However, there is currently no research mentioning their association and more research is needed to confirm them.

### 4.4. Predictive Model for Macrosomia

In this study, a prediction model for macrosomia was first constructed based on different species between the two groups. The AUC of the model was 0.907 (95% CI: 0.816–0.963), which showed good predictive performance. On the basis of this model, relevant clinical indicators were added to predict the AUC of the model, which increased to 0.935 (95% CI: 0.851–0.979). Meanwhile, the importance of differential species in the model is higher than clinical indicators, once again indicating a close connection between the differential species we have found and the occurrence of macrosomia.

At present, most of the prediction models for macrosomia were based on clinical indicators and maternal characteristics and the AUC of the model was in the range of 0.71–0.91 [51]; few predictive models involved gut microbiomes. Two-dimensional ultrasound is often used to predict the birth weight of fetuses but its sensitivity and specificity are only 0.56 (95% CI 0.49–0.61) and its predictive efficacy is not high [52]. Compared with two-dimensional ultrasound, three-dimensional ultrasound and magnetic resonance imaging have higher predictive accuracy and are also used to evaluate fetal weight [53] but their examination prices are more expensive and may have side effects. Mazouni et al. developed a column chart model that combines clinical and ultrasound data to predict macrosomia. The model has good discrimination and correction capabilities but it is only applicable to Europeans and Africans, not Chinese people [54]. Zou et al. constructed a column chart model for predicting macrosomia in China based on pre-pregnancy BMI, gestational weight gain, fasting blood glucose, TG, biparietal diameter, and amniotic fluid index. The AUC of the model was 0.903 and the validation results were all above 0.8, indicating good predictive effects. However, the collection of indicators included in these models must continue until late pregnancy, so it is not possible to predict macrosomia in early pregnancy [55], which has no practical significance for the prevention of macrosomia. Compared with previous studies, our study included different gut microbiomes and five common indicators of early pregnancy to construct a prediction model for macrosomia. The AUC reached 0.935, indicating good predictive performance. The random forest model using clinical and gut microbiota targeting biomarkers has shown effective performance in predicting macrosomia. However, considering the impact of race and region on gut microbiota, it is necessary to validate the model with different populations to evaluate whether gut microbiota is suitable for predicting macrosomia in early pregnancy among populations of different regions and ethnic groups.

### 4.5. Strengths and Limitations

Compared with previous reports, our study discovered some new gut microbiomes that may affect newborn birth weight. In addition, we found the relative pathway of macrosomia and combined it with the taxonomic results to explain the possible mechanism underlining the incidence of macrosomia. Moreover, we first constructed a predictive model including the gut microbiome to predict macrosomia and showed good predictive value.

It is worth noting that our study was limited by a relatively insufficient sample size and further studies with larger numbers of subjects are needed. Moreover, we did not collect and sequence fecal samples from the second and third trimesters to identify whether the gut microbiome varied during different periods of pregnancy. We also did not collect stool samples of newborns. So, we cannot directly prove whether the intestinal microbes of the mother exist in the fetus. Further research can be conducted in this area. Although a relationship between the maternal gut microbiome and macrosomia was observed, more research is needed to examine and confirm the discovered microbiome.

## 5. Conclusions

In summary, we found there was a correlation between the maternal gut microbiome and macrosomia and identified some gut microbiota like *Bacteroides salyersiae*, *Bacteroides plebeius*, *Ruminococcus lactaris*, and *Bacteroides ovatus* that may influence the occurrence of macrosomia. Novel associations were found between the differential gut microbiome and pathway of gondoate biosynthesis, L-histidine degradation III, cis-vaccenate biosynthesis, and mannitol cycle. We speculated that *Bacteroides salyersiae*, *Bacteroides plebeius*, and *Bacteroides ovatus* may affect the functions of gondoate biosynthesis and cis-vaccenate biosynthesis or secrete SCFAs and thus result in related TGs and glucose. More TGs and glucose may flow to the fetus and influence fetal development. Further well-designed studies are required to investigate and examine these microbiomes. Early prediction models based on differential gut microbiota and differential clinical indicators have good predictive ability for the occurrence of macrosomia. Altogether, these findings open up new strategies for the prevention of macrosomia through modulation of the intestinal microbiota.

## Figures and Tables

**Figure 1 biology-13-00570-f001:**
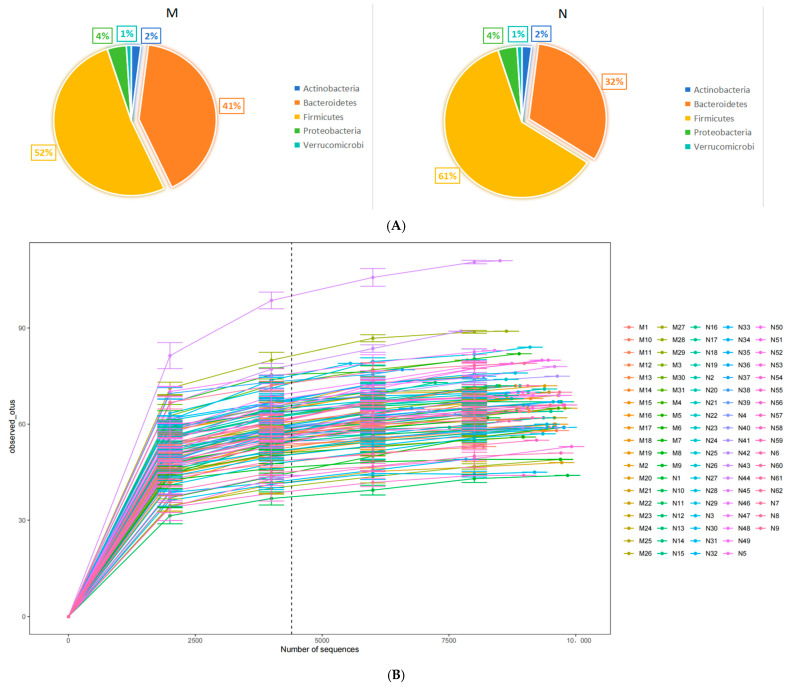
The comparison of the gut microbiome between two groups. (**A**) Pie charts of phylum composition between two groups (**M**: cases and **N**: controls). (**B**) Rarefaction curve of the observed species.

**Figure 2 biology-13-00570-f002:**
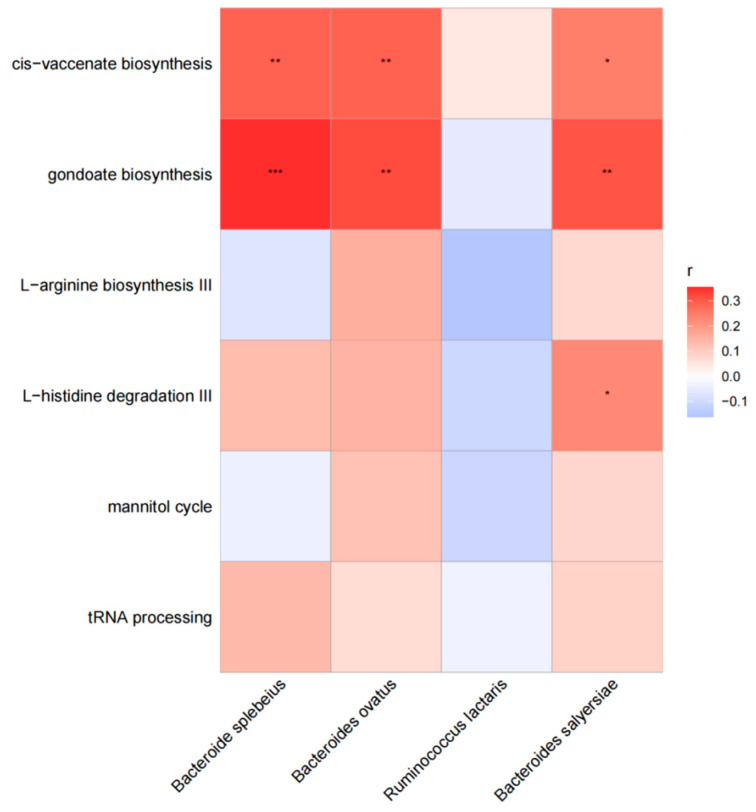
The association between different functional pathways and different species in cases. The horizontal are different functional pathways, and the longitudinal are different species. The grid values represent Spearman’s rank correlation coefficient (r), ranging from −1 to 1. Negative correlations (r < 0) are blue, positive correlations (r > 0) are red. Significance is indicated by * for *p* < 0.05, ** for 0.001 < *p* ≤ 0.01, and *** for *p* ≤ 0.001.

**Figure 3 biology-13-00570-f003:**
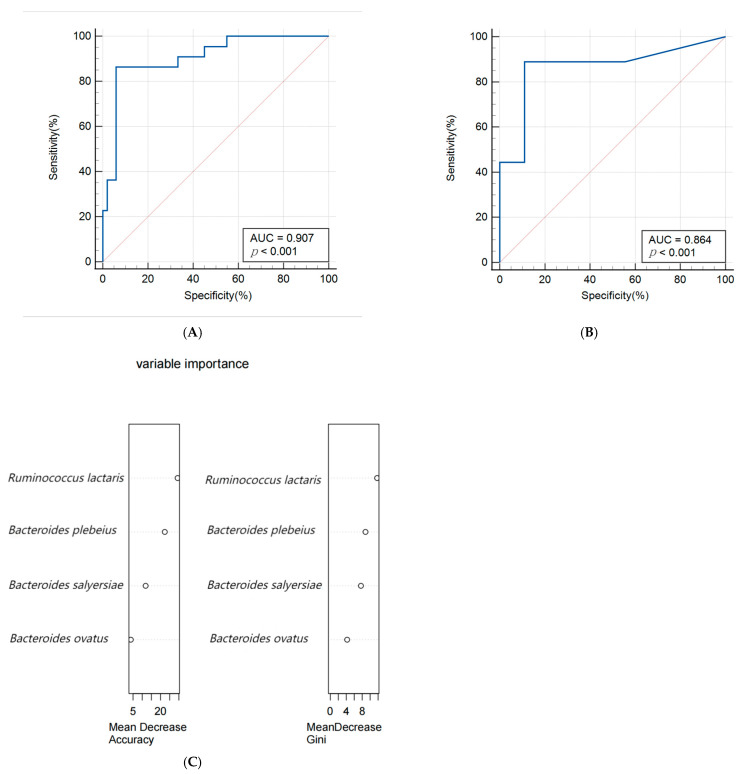
(**A**,**B**) Receiver operating characteristic (ROC) curve of the predictive model for macrosomia using different species in the 74 samples of the training set (**A**) and 19 samples of the validation set (**B**). (**C**) The variable importance of this model.

**Figure 4 biology-13-00570-f004:**
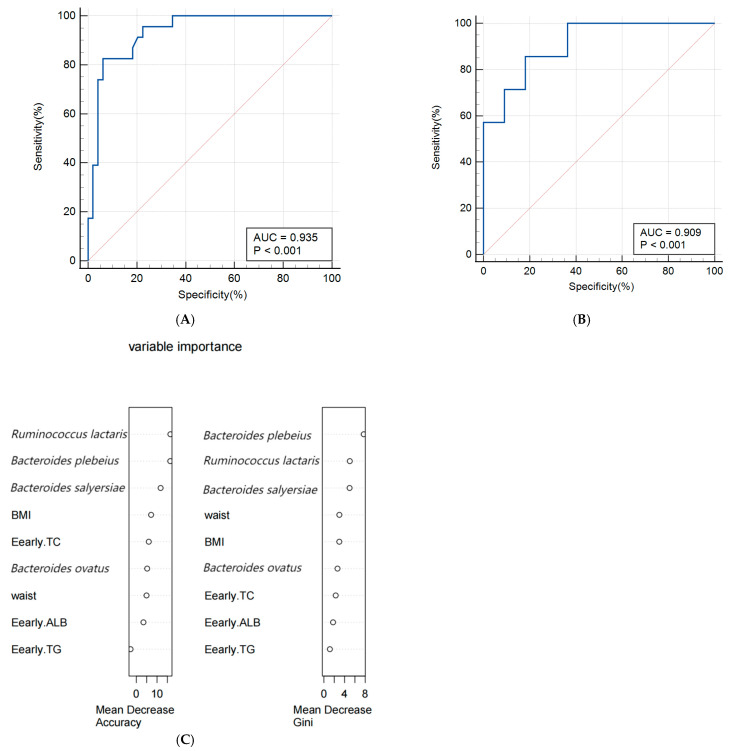
(**A**,**B**) Receiver operating characteristic (ROC) curve of the predictive model for macrosomia using different species and clinical indicators in the 74 samples of training set (**A**) and 19 samples of validation set (**B**). (**C**) The variable importance of this model.

**Table 1 biology-13-00570-t001:** Basic characteristics of participants.

	Cases (*n* = 31)	Controls (*n* = 62)	*p*
**Basic characteristics and Anthropometrics indicators in early pregnancy**			
Age of delivery (years)	30.58 ± 3.70	30.66 ± 3.83	0.923
Waist (cm)	82.28 ± 6.28	77.35 ± 7.21	0.002 *
Weight (kg)	58.69 ± 7.60	52.31 ± 7.69	<0.001
Hight (cm)	160.19 ± 4.08	158.52 ± 4.16	0.07
Body Mass Index	22.87 ± 2.88	20.77 ± 2.57	0.001
SBP (mmHg)	115.32 ± 11.41	115.80 ± 10.38	0.838
DBP (mmHg)	74.65 ± 8.34	75.09 ± 8.62	0.810
**History of diseases**			
Gestational Hypertension	2 (6.45%)	1 (1.61%)	0.257
Gestational Diabetes Mellitus	0 (0.00%)	0 (0.00)	1.000
Adverse Pregnancy	19 (61.29%)	38 (61.29%)	1.000
**Biochemical parameters in the first trimester**			
HGB (g/L)	126.67 ± 8.95	125.37 ± 8.44	0.503
GLU (mmol/L)	4.74 ± 0.31	4.64 ± 0.35	0.172
ALB (g/L)	44.79 ± 2.86	45.98 ± 2.58	0.045
TG (mmol/L)	1.76 ± 0.72	1.47 ± 0.46	0.019
TC (mmol/L)	4.89 ± 0.71	4.49 ± 0.71	0.012
HDL-C (mmol/L)	1.87 ± 0.40	1.95 ± 0.32	0.274
LDL-C (mmol/L)	2.70 ± 0.59	2.42 ± 0.69	0.060
AST (U/L)	16.40 (6.40)	18.10 (9.00)	0.476
ALT (U/L)	14.60 (10.50)	17.95 (15.40)	0.471
**Eating habits**			
Daily cereals intake (g)	65.57 ± 26.012	69.29 ± 26.07	0.522
Daily tubers intake (g)	50.72 ± 44.64	45.76 ± 31.27	0.538
Daily vegetables intake (g)	116.85 ± 54.28	110.94 ± 54.70	0.627
Daily fruits intake (g)	164.19 ± 66.86	165.85 ± 55.02	0.900
Daily meat intake (g)	35.75 ± 30.84	31.88 ± 22.53	0.497
Daily eggs intake (g)	35.80 ± 14.73	29.77 ± 13.88	0.059
Daily milk intake (mL)	41,280.13 ± 79.27	25,648.51 ± 35.70	0.193
Daily soybeans intake (g)	48.33 ± 43.25	48.65 ± 42.58	0.973
Daily peanuts intake (g)	42.77 ± 28.87	28.48 ± 20.66	0.008
Daily oil intake (g)	10.35 ± 1.35	9.99 ± 1.41	0.256
Daily salt intake (g)	5.10 ± 1.39	4.69 ± 0.41	0.032
Dail water intake (mL)	490.28 ± 251.93	484.90 ± 214.29	0.915
**Mode of delivery**			0.002
Vaginally delivery	6 (19.40%)	36 (58.10%)	
Assisted vaginal delivery	1 (3.20%)	2 (3.20%)	
Cesarean section delivery	24 (77.40%)	24 (38.70%)	
**Obstetric history**			
Gravidity			0.056
1	8 (25.80%)	28 (45.16%)	
≥2	23 (74.19%)	34 (54.84%)	
Parity			0.206
0	16 (51.61%)	39 (62.90%)	
≥1	15 (48.39%)	23 (37.10%)	
Abortion			0.587
No	12 (38.71%)	24 (38.71%)	
Yes	19 (61.29%)	38 (61.29%)	
**Basic characteristics of newborns**			
Sex			0.518
Male	19 (61.30%)	38 (61.30%)	
Female	12 (38.70%)	34 (54.80%)	
Birth weight (g)	4183.87 ± 194.80	3005.65 ± 567.36	<0.001
Premature labor	0 (0.00%)	3 (4.80%)	0.548

Significance is indicated by * for *p* < 0.05.

**Table 2 biology-13-00570-t002:** Different species in Maaslin2.

Species	Coefficient	*p*	Q	Coefficient *	*p* *	Q *
*Bacteroides salyersiae*	−6.897	<0.001	<0.001	−6.748	<0.001	<0.001
*Bacteroides plebeius*	−5.210	<0.001	<0.001	−5.049	<0.001	<0.001
*Ruminococcus lactaris*	−4.994	<0.001	<0.001	−4.880	<0.001	<0.001
*Bacteroides ovatus*	−2.013	0.003	0.127	−1.909	0.004	0.183

* Coefficient, Q, *p* value after adjusting for eating habits including average daily intake of salt and peanut, BMI, waist.

**Table 3 biology-13-00570-t003:** Different pathways in Maaslin2.

Pathway	Coefficient	*p*	Q	Coefficient *	*p* *	Q *
mannitol cycle	−2.321	<0.001	0.002	−2.156	<0.001	0.003
L-arginine biosynthesis III	−1.363	<0.001	0.011	−1.213	0.001	0.011
tRNA processing	−2.643	<0.001	0.011	−2.482	0.001	0.011
L-histidine degradation III	−1.274	<0.001	0.014	−1.124	0.001	0.011
gondoate biosynthesis	−0.913	0.002	0.131	−0.763	0.001	0.121
cis-vaccenate biosynthesis	−0.805	0.004	0.223	−0.655	0.004	0.022
thiamin salvage II	−0.840	0.005	0.231	−0.689	0.005	0.258

* Coefficient, Q, *p* value after adjusting for eating habits including average daily intake of salt and peanut, BMI, waist.

## Data Availability

The raw sequence data reported in this paper were deposited in the Genome Sequence Archive in BIG Data Center, Beijing Institute of Genomics (BIG), Chinese Academy of Sciences, under accession numbers CRA007116, the shared URL is https://ngdc.cncb.ac.cn/gsa/s/Y7LcV2Kh (4 June 2022).

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
