# Peer review of "Association between the Maternal Gut Microbiome and Macrosomia"

_biology, 2024, doi:10.3390/biology13080570_

Round 1

Reviewer 1 Report

Comments and Suggestions for Authors

This manuscript studied maternal gut microbiome in the patients with fetal macrosomia. Maternal gut microbiome change during pregnancy is a hot topic in the maternal-fetal care field. The gut microbiome change reflexes the physiological change in the mothers and affects the fetal growth. And the references are appropriate.

Weakness of the study:

1. Several confounding variables did not reported in the study which might draw the biased conclusion. For example, GDM status (only history of GDM reported), father's weight and height (genetic variable), GTPAL and et al.

2. The predictive model did not remove confounding variables, like BMI, weight. 

3. Figure1 pie chart is not clear presented. 

4. The result is not well-written.

Comments on the Quality of English Language

Need to improve, especially result section

Author Response

Comments 1:Several confounding variables did not reported in the study which might draw the biased conclusion. For example, GDM status (only history of GDM reported), father's weight and height (genetic variable), GTPAL and et al.

Response 1:Thank you very much for your strong support of our work!We agree with the your suggestions and will incorporate the recommended changes into the manuscript.

①In response to the comment about the paper, we have reported the results of GTPAL. The results appeared fairly stable across the different groups(Table 1. Obstetric history), and therefore are unlikely to have confounded our results.

② We apologize for the lack of information about father's weight and height. This oversight occurred during the design of the questionnaire and data collection process, as confounding variables were not adequately considered.

③We only reported GDM history in this study due to the collection of indicators in the first trimester, during which time the status of GDM was not definitively determined until the second trimester.

Comments 2:The predictive model did not remove confounding variables, like BMI, weight.

Response 2:In the initial results of variable importance in this model (Figure 4-C), weight was not considered among the included variables in the predictive model. However, weight was mistakenly included in the text as a result of an oversight. We regret this error and have rectified it in the revised version of the manuscript. (Line 286-287, page 9).

Comments 3:Figure1 pie chart is not clear presented.

Response 3:We have revised the figure to enhance readability (figure 1), and we appreciate the valuable suggestion.

Comments 4: The result is not well-written.

Response 4: Thank you for your suggestion. We do invite a friend of us who is a native English speaker from the USA to help polish our article. And we tried our best to improve the manuscript. All the changes have been marked in yellow in the revised paper(line 200-215, line 218-228, line 262-268, line 274-277), and we hope the correction will meet with approval. Additionally, If there are any other modifications we could make, we would like very much to modify them and we really appreciate your help.

Reviewer 2 Report

Comments and Suggestions for Authors

The manuscript entitled "Association between maternal gut microbiome and macrosomia" aims to identify elements of the maternal microbiome that may be related to macrosomia in infants, identify the mechanism and how these microbiome could contribute to macrosomia and develop a prediction model for macrosomia in infants based on the gut microbiome of mothers in the first trimester. Each of these 3 objectives was successfully achieved in the study where the authors identified 4 microbial species, three bacteriodes species and a ruminococcus species to be present in the first trimester microbiome of infants with macrosomia. The metabolic activities of these microorganisms were interrogated and correlations made in relation to cis-vaccenate biosynthesis and gondoate biosynthesis and how metabolites could affect maternal blood lipid content. The authors also developed and tested a prediction model for macrosomia based on 5 indicators from the early pregnancy gut microbiome.

While the authors are clear that the sample size was a limitation of the study it is not clear how the control group were selected from the 920 mothers that had normal birth weight infants.  Overall This represents a nice observational study but further research would need to be conducted with larger numbers.

The manuscript is well written. for clarification line 42 page 1 could be amended to : "....compared with mothers bearing fetuses that had anormal birth weight"....

Author Response

Comments1: The manuscript is well written. for clarification line 42 page 1 could be amended to : "....compared with mothers bearing fetuses that had normal birth weight"....

Response1: We sincerely thank the reviewer for careful reading. As suggested by the reviewer, we have corrected the “...compared with fetuses with normal birth weight” into “...compared with  mothers bearing fetuses that had normal birth weight. ”(line 43-44, page 1). And we tried our best to polish the manuscript. All the changes have been marked in yellow in the revised paper(line 200-215, line 218-228, line 262-268, line 274-277), and we hope the correction will meet with approval. Additionally, If there are any other modifications we could make, we would like very much to modify them and we really appreciate your help.